

# SPREAD: A high-resolution daily gridded precipitation dataset for Spain

Roberto Serrano-Notivoli[1,2], Santiago Beguería[2], Miguel Ángel Saz[1], Luis Alberto Longares[1] and Martín de Luis[1]

[1] Department of Geography and Regional Planning and Environmental Sciences Institute (IUCA), University of Zaragoza, Zaragoza, E-50009, Spain.
[2] Estación Experimental de Aula Dei, Consejo Superior de Investigaciones Científicas (EEAD-CSIC), Zaragoza, E-50059, Spain.

*Correspondence to*: Roberto Serrano-Notivoli (rs@unizar.es)

**Abstract.** A high-resolution daily gridded precipitation dataset was built from raw data of 12,858 observatories covering a period from 1950 to 2012 in peninsular Spain and 1971 to 2012 in Balearic and Canary Islands. The original data were quality controlled and gaps were filled on each day and location independently. Using the serially-complete dataset, a grid with a 5x5 kilometres spatial resolution was constructed by estimating daily precipitation amounts and their corresponding uncertainty at each grid node. Daily precipitation estimations were compared to original observations to assess the quality of the gridded dataset. Four daily precipitation indices were computed to characterize the spatial distribution of daily precipitation and nine extreme precipitation indices were used to describe the frequency and intensity of extreme precipitation events. The use of the total available data in Spain, the independent estimation of precipitation for each day and the high spatial resolution of the grid allowed for a precise spatial and temporal assessment of daily precipitation that are difficult to achieve when using other methods, pre-selected long-term stations or global gridded datasets. SPREAD dataset is publicly available at http://dx.doi.org/10.20350/digitalCSIC/7393.

## 1 Introduction

Daily precipitation is a key variable to understanding the behaviour of extreme weather events and the severe impacts they cause on hydrological systems, in natural systems and human societies. These impacts can be considered in regional and local plans which can help to mitigate major disasters if the correct environmental data are available. Unfortunately, the raw climatic information is usually highly fragmented in time and discontinuous in space. Quality control and reconstruction processes are therefore in high demand, as well as final products such as serially-continuous observational series or gridded datasets. High-resolution spatiotemporal precipitation datasets are useful tools for land management, and the analysis of the



spatial patterns of daily precipitation needs a complete coverage of the study area that only high-resolution gridded data can provide.

The number and density of climate observatories are finite. Therefore, there have been many contributions over the last decade to solve this problem by creating high-resolution and world-based gridded datasets based on the (geo)statistical

relations of the observatories (New et al., 2002; Hijmans et al., 2005; Di Luzio et al., 2008; Harris et al., 2014; Schamm et al., 2014). However, these global datasets are often not optimum for regional analysis, where a higher resolution both in space and time is required (Herrera et al., 2012). Increasing temporal resolution from monthly or annual to daily scale allows analysis of other relevant components of the climate, such as the extreme precipitation events or the length and intensity of dry spells that preceed most of the annual hydrological processes. In this respect, daily precipitation grids of different spatial

resolutions have been developed at the global (Piper and Stewart, 1996; Menne et al., 2012) or regional scale (Frei and Schär, 1998; Eischeid et al., 2000; Kyriakidis et al., 2001; Rubel and Hantel, 2001; Klein-Tank et al., 2002; Hewitson and Crane, 2005; Liebmann and Allured, 2005; Haylock et al., 2008; Klok and Klein-Tank, 2009; Mekis and Vincent, 2011; Hwang et al., 2012; Yatagai et al., 2012; Jones et al., 2013; Chaney et al., 2014; Isotta et al., 2014; Hernández et al., 2016).

Also a few daily datasets have been made for Spain or some of its regions (Brunet et al., 2006; Vicente-Serrano et al., 2010;

Herrera et al., 2012). These datasets are useful to analyse global precipitation, which in Spain have a wide range of different spatial and temporal distributions (Esteban-Parra et al., 1998; Rodríguez-Puebla et al., 1998; Belo-Pereira et al., 2011; De Luis et al., 2011; González-Hidalgo et al., 2011; Cortesi et al., 2014). But they are also suitable to analyse spatial patterns in daily characteristics of precipitation through extreme indices that allow comparisons between regions. In Spain, these indices have been applied to mainland Spain (Martín-Vide, 2004; Herrera et al., 2012; Merino et al., 2015) and to specific regions

(Casas et al, 2007; Martínez et al., 2007; Rodrigo and Trigo, 2007; Lopez-Moreno et al., 2010). All of these datasets used the precipitation estimations to compute climatic indices. However, unlike temperature, there is more uncertainty in precipitation estimation due to its higher temporal and spatial variability, especially at the daily time scale. Gridded datasets have an uncertainty that has to be measured and addressed depending on the method used to predict daily precipitation. The uncertainties of these estimations depend on the density of observations used to compute the model (Hofstra et al., 2010). As

the density decreases, uncertainties increase, demanding more complex models (Tveito et al., 2008). This issue has important

implications in subsequent climatic analyses, and this is one consideration that needs to be taken into account (Beguería et al., 2015).

In this paper, we present SPREAD (Spanish PREcipitation At Daily scale), a new high-resolution daily gridded precipitation

dataset for Spain. Thirteen daily and extreme precipitation indices were calculated as an example of applicability, characterizing daily precipitation distribution (four indices to characterize the spatial distribution of daily precipitation and nine for extreme precipitation). The grid was computed for 1950-2012 period covering peninsular Spain and 1971-2012 for Balearic and Canary Islands. It was built from 12,858 original stations by creating reference values (RV) using Generalized Linear Models (GLM) based on the 10 nearest observations using altitude, longitude and latitude as covariates. All the

calculations were developed with *reddPrec* (https://cran.r-project.org/web/packages/reddPrec), an R package containing the required functions to make reconstruct original daily precipitation series and create grids (Serrano-Notivoli et al., 2017).

This paper is organised as follows: Section 2 describes the original dataset. Section 3 shows the methods applied for the reconstruction and the gridding process considering uncertainties of estimates. Also, the statistical procedures are explained here. Section 4 shows the results of the reconstruction, gridding procedure and spatial distribution of daily and extreme

precipitation characteristics in Spain. Section 5 specifies the availability of the dataset. The results are discussed at Section 6 with some final remarks in Section 7.

## 2 Input data

A total of 12,858 observatories covering Spain were used. The grid was divided in three areas: 1) Peninsular Spain with 11,513 stations covering the period 1950-2012; 2) Balearic Islands, with 425 stations covering 1971-2012 and 3) Canary

Islands covered 1971-2012, using 920 stations (Figure 1 down).

Most of the data were provided by the Spanish Meteorological Agency (AEMET), but we also used data from regional hydrological and meteorological services, and from the national agronomic network (Table 1). The mean length of the original series was 18.8 years, and only 17 of the 12,858 original observatories had covered the period 1950-2012. However,

the spatial distribution of the observatories showed a remarkable density (Figure 1 main map), which is useful to make proper reconstructions.

## 3 Methods

### 3.1 The reference values (RV) as a general tool for quality control and reconstruction

The key process for climate reconstruction is based on the calculation of individual reference values (RV) for each day and location, according to the information available from the closest neighbouring stations. Generalized linear models (GLM) are used to compute the RV using the precipitation data (occurrence and magnitude) of the ten nearest neighbours as the dependent variable and the geographic information of each station (latitude, longitude and altitude) as the independent variables.

The geographic information of each station (latitude, longitude and altitude) is used as the independent variable for modelling. As the data availability varies from day to day, selected neighbour stations also vary. Since independent models are constructed for each location and day, the estimated parameters of the models (reflecting the influence of the independent variables and their probability to influence the occurrence and magnitude of precipitation), may also vary with measurement day and location.

Including three factors in the model allows for increased sensitivity of the model to be able to reflect the local changes in precipitation patterns. Because this method is based on local and independent data points across time, there are no restrictions imposed due to the length or structural characteristics of the series allowing efficient use of all of the available information.

The computation of each individual RV is based on two predicted values: (i) a binomial prediction (BP) of the probability of

occurrence of a wet day; and (ii) a magnitude prediction of precipitation (MP), in the case where a wet day is predicted. The combination of these two values (RV=MP if BP>0.5, else RV=0) produce the estimated (RV) and its corresponding standard error for each day and location. Further details on the statistical procedures are described and discussed in Serrano-Notivoli et al., (2017).

The obtained RV were first used to develop a quality control (QC) test in order to detect anomalous data in the original dataset, and then to estimate precipitation at all missing locations and days for the whole dataset. Finally, a 5 x 5 kilometre grid was built for all the whole of Spain and the entire period of reconstructed stations. The reconstruction and gridding processes were applied using the R package *reddPrec* following the methodology described in Serrano-Notivoli et al. (2017).

The QC process detected and removed suspect data by comparing daily values registered at each station with predicted values calculated from its surrounding observations. Five criteria were used to flag and remove suspect data from the original dataset: 1) suspect data: observed value was over zero and all their 10 nearest observations were zero; 2) suspect zero: observed value was zero and all its 10 nearest observations were over zero; 3) suspect outlier: the magnitude of the observed value was 10 times higher or lower than that predicted by its 10 nearest observations; 4) suspect wet day: observed

value was zero, wet probability was over 99%, and predicted magnitude was over 5 mm; and 5) suspect dry day: observed value was over 5 mm, dry probability was over 99%, and predicted magnitude was under 0.1 mm.

Once the QC was completed, a new set of RV were calculated using the curated dataset. Since the RVs were calculated for all days and locations, including those for which an observation exists but without using that observation, the comparison between RVs and the corresponding observed values constitutes a leave-one-out cross-validation (LOO-CV) process. A

number of goodness of fit statistics were therefore used for assessing the quality of the estimated values (RVs): the mean absolute error (MAE), as a measure of the error magnitude; the mean error (ME) and the ratio of means (RM = mean of estimations / mean of observations), as a measure of bias; and the ratio of the standard deviations (RSD = std. dev. of estimations / std. dev. of observations), as a measure of bias in the variance. These statistics were computed for monthly aggregates and for 13 daily and extreme precipitation indices (described on section 3.3).

Additionally, scatterplots were made between observations and estimations of daily precipitation at the station locations using the number of zero precipitation days (dry days); daily means (considering all days); medians of the wet days; the 95[th] percentile of wet days, and the Pearson's correlation statistic was computed in each case.

Finally, the missing values in the original data series were filled with the reference values (RV). From 12,858 original observatories, we reconstructed those that had more than 10 years of original data (7,604 stations). This guaranteed a reliable

reconstructed series with enough observations to compute the grid.

### 3.2 Gridding and uncertainty

The same procedure based on the calculation of RV was used to build a 5 x 5 km spatial resolution grid. For each point of the grid (*x, y and z*) and each day of the total period, RV was computed based on the data of the ten closest reconstructed stations.

As a measure of uncertainty, we computed the standard error (in mm) for each RV. Using the ratio between this error and the RV we obtained the relative error (expressed in percentage) in each index and aggregation (monthly, annual and seasonal).

### 3.3 Applications: Daily mean and extreme precipitation indices

As examples of possible applications of the gridded dataset, four indices using daily precipitation were calculated to characterize the spatial distribution of daily precipitation and its extremes (nine more indices) (Table 2). Most of these
indices are included in the suite of extreme precipitation and temperature indices (Zhang et al., 2011) developed by WMO Expert Team on Climate Change Detection and Indices (ETCCDI). They have been applied in previous works to assess the distribution of extreme events in many areas (e.g. Donat et al., 2013 and 2014; Keggenhof et al., 2014; Asadieh and Krakauer, 2015; Sanago et al., 2015; Yin et al., 2015; Sigdel and Ma, 2016). All the indices were computed at the annual scale and the average annual values were calculated.

## 4 Results

### 4.1 Reconstruction of the observational dataset and grids

The quality control process flagged and removed an annual average of 2.4% of data in the peninsular Spain, 1.7% in the Balearic Islands and 1.8% in the Canary Islands. There were no major differences in the number of removed data by years. A brief increase was observed in the first 20 years in peninsular Spain data (Figure 2 a), while from 1971 to the end of the
period the number of removed data barely changed.

In the Balearic Islands the number of removed data was more variable (Figure 2 b). Suspect data and zeros were usually detected because they represented low precipitation values with estimates from 0 to 1 or 2 mm, so they were included in one of these two criteria. The low detected values of outliers, suspect dry and wet data were probably due to the configuration of the islands, small with a high density of observatories even at high elevations. In the Canary Islands, although the dataset

contained more data series than in the Balearic Islands, and the number of available daily stations was very variable, the

quality control process removed a similar number of data over the collection period (Figure 2 c). This was also the area with

the least number of suspect zeros of the three localities.

A complete 5 x 5 km spatial resolution grid was calculated based on the reconstructed station series. Daily precipitation was

estimated from 1950 to 2012 in the peninsular Spain and from 1971 to 2012 in Balearic and Canary Islands. The standard

error of the model used to compute the estimations was calculated as a measure of uncertainty for each day and grid point.

**4.2 Observations – estimations comparison**

Daily precipitation values were estimated at the same locations and days as the observed dataset for comparison purposes.

This section shows the results of this comparison.

**Wet/dry estimation**

The number of observed zero precipitation days (dry days) in the entire study area was 57,761,815 and the estimated (only

for corresponding days with observations) was 57,773,250 (a ratio of 0.9998), so it can be concluded that this method is not

biased in the prediction of wet / dry days. The comparison between the original dataset and the corresponding estimates

showed a high correlation in both the Spain peninsula, Balearic Islands and Canary Islands (Pearson 0.83, 0.85 and 0.73

respectively), with similar frequency distribution by station (see supplementary material figure S1). Terming the wet days as

positive (observed P >1) and the dry days as negative (observed P = 0), the true negative rate p (RV=0|P=0) was over 94% in

all cases (Table 3), and the true positive rate or precision p (RV>0|P>0) was over 79% except in the Canary Islands, where it

decreased to 70%. To a large part, the false negatives p (RV=0|P>0) and false positives p (RV>0|P=0) were due to the

prediction of precipitation in days with low amounts. In events with very low precipitation amounts, the estimate of the

probability of occurrence was likely to be dry, despite the fact that the station could register a minimum quantity of rain

(usually under 1 or 2 mm). This causes a brief difference in amounts, becoming more distinct in the dry/wet accuracy

assessment.



**Magnitude estimation**

The comparison between amounts of observed and estimated precipitation showed a high correlation both in daily means (daily mean precipitation by stations, considering the whole series), daily medians in wet days (only considering days with P > 0 in observations and estimations) and in the 95th percentile of wet days (Figure 3). Daily precipitation means (considering

dry and wet days) reached the maximum correlation between observations and estimations (Figure 3a, d, g), decreasing in daily precipitation medians on wet days (P>0) (Figure 3b, e, h) and considering only the daily precipitation over the 95th percentile on wet days (Figure 3c, f and i). However, in all cases the values of the Pearson correlation coefficient were over 0.93. In the Canary Islands the goodness of fit was lower than in the other areas. The Canary Islands experience high orographic rainfall and climatic variability, which is thought to have contributed to the lower availability of data. Most of

these islands have their own climate, with high differences between both sides (North-South in eastern islands and East-West in western islands). As the number of observatories on the Canary Islands was limited and the estimates were from a low density of stations, a greater radius was used to get the minimum number of stations required to run the model. This can lead to a more inaccurate estimation, although, the aggregation by days instead of stations showed a better agreement, with correlations over 0.96 in most of the cases (see supplementary material figure S2).

The histograms of estimated and observed precipitation (Figure 4) showed a good general agreement. There was a slight over-estimation of the values below 1 mm, and a slightly flatter distribution around the mean. The agreement between the histograms was high above 10 to 20 mm. The differences found in the lowest values (under 0.1 mm) were due to a large part to the fact that estimates below 0.1 mm were allowed. This value was the minimum measurement in the observed dataset. These results were similar in the peninsular Spain, the Balearic Islands and the Canary Islands datasets.

One common problem to most observational datasets is the uneven distribution of the stations with respect to the altitude. Overall, the high elevation areas tend to be under represented in the datasets due to a lower spatial density of stations. This could result in biases in any derived datasets. For instance, in Spain only around 2% of the stations are above 1,500 m. a.s.l., which represents 4% of the Spanish territory. For this reason, it is relevant to evaluate the goodness of fit of the estimated values by altitudinal ranges (Table 4). The RM showed that there were no substantial biases (values close to 1) until 1,500 m.

a.s.l., and only a slight overestimation of 6-7% above this altitude in peninsular Spain. In the Balearic Islands there were

under and over-estimations above 500 m. a.s.l., but the number of stations is too low to consider these values representative. In the Canary Islands precipitation was slightly underestimated (-8%) at higher altitudes (> 2,000 m. a.s.l.).

Assessing the monthly aggregates of daily precipitation (Table 5), the results were very similar to the ratio of means (RM), indicating the absence of systematic biases, with the exception of November and December which showed a slight under-
estimation. The ratio of the standard deviations (RSD) was also very close to one in the peninsular Spain and the Balearic Islands, indicating no biases in the variance estimation, although there was a small under-estimation in November and December. The RSD was more variable in the Canary Islands, with an over-estimation between 10 and 20% in the summer (June to August), and underestimation of around 10% in November and December. Very low values of MAE were also found (average of 9.89, 8.98 and 5.99 in peninsular Spain, the Balearic and Canary Islands, respectively), as well as ME
(average of -0.41, -0.29 and -0.73). These months usually receive low precipitation (most of them zero) and the events are typically of small spatial extend, leading to higher uncertainty of the estimations, thus producing low values of fit between observed and estimated precipitation. However, the RM was near to 1 in all cases, indicating the absence of bias in the estimations despite the variable uncertainty.

The development of a spatially and temporally complete gridded dataset allowed the assessment of the characteristics of
daily precipitation over Spain. For that, 13 daily and extreme precipitation indices were computed. The comparison between the observed and the estimated values of the indices showed values of RM and RSD near to 1 in all the considered spatial units and indices (Table 6), indicating no substantial biases in the mean and variance of the estimated indices. However, RX1 showed a slight under-estimation in peninsular Spain and the Balearic Islands, as so did R20mm in peninsular Spain and SDII in the Canary Islands, where the number of wet days (NWD) was over-estimated. Overall, all the indices were similar
at observed stations and their corresponding estimates in the Canary Islands showed the largest differences, as shown at the monthly scale (Table 5).

## 4.3. Spatial distribution and uncertainty in daily precipitation

The daily mean precipitation intensity (PMED) (computed as the median precipitation in wet days) map (Figure 5 a) showed three areas with maximum values (Central Range, Pyrenees and Betic Range). Overall, PMED was higher (>8 mm) in the

south-western sector of peninsular Spain, in the north-west and in the Pyrenees. The rest of the territory reached values between 4 and 6 mm.

The NWD showed a strong gradient from the northwest (Figure 5 c), with more than 100 days of precipitation, to the south-east, where the lowest was less than 30 days of precipitation per year. This value was the most frequent in the Canary Islands

except in the La Palma island (extreme west) and in the highest areas of the other islands (over 1,500 m. a.s.l), where rainfall occurrence reached almost 90 days per year. The rest of peninsular Spain and the Balearic Islands showed average values between 50 and 70 days.

The mean length of the dry spells (CDDm) (Figure 5 e) showed a similar gradient. The northern sector reached values under 5 days, increasing to the southeast where the average length of consecutive dry days was more than 30 days, like in the

largest part of the Canary Islands.

NWD, CDDm and mean consecutive wet days (CWDm) characterized daily precipitation frequency, showing that in the southwest of peninsular Spain and the Canary Islands the average number of rainy days per year is less than 30 and, when rain occurs, the mean length of the events was less than 1.5 days. Conversely, the mean number of wet days in almost the entire Cantabric fringe was over 120 days per year, and the mean consecutive number of dry days was under 5 days.

The uncertainty of estimates in daily precipitation indices was spatially variable, but in all cases it had an increasing gradient from north-west to south-east in peninsular Spain, especially in PMED and NWD. This uncertainty, which was not necessarily similar to the distribution of its corresponding variable, informed about the reliability of the results of the indices. The higher values in most of the indices occurred in the south-west.

**4.4. Spatial distribution and uncertainty in daily extreme precipitation**

Mean precipitation in wet days (SDII) (computed as the mean precipitation in wet days) in Spain ranged between 5 and more than 25 mm (Figure 6). The lowest values were distributed in the Northern and Southern plateaus and at the bottom of the Ebro Valley, unlike the south-east of peninsular Spain which had higher values similar to the eastern coast, where the total rainfall was low but daily precipitation intensity was very high. The normal values in the Pyrenees were higher than 15 mm for each wet day, especially in the western half where the Atlantic influence is strongest. Despite this, the highest values

were in the Central Range, with more than 25 mm per wet day, being the area with most precipitation on wet days, in the whole country.

The mean maximum precipitation in one day (RX1) (Figure 6) was concentrated mainly in the highest areas, which create orographic barriers. The Central Range was the only zone in Spain that reached values higher than 200 mm, decreasing with

elevation until 40 mm. This pattern was replicated in most of the high-elevation areas of peninsular Spain and islands, but also in the Mediterranean coast, which is characterized by a high frequency of extreme events. The distribution of the maximum precipitation in five days (RX5) (Figure 6) was very similar, but with a smoother gradient. As the variability in the extreme nature of the RX5 was less intense, the spatial distribution was more homogeneous with softer differences between the regions.

A high number of days per year with more than 10 mm of precipitation (R10mm, Figure 6) is relatively frequent in Spain, and especially so in a Mediterranean climate as it corresponds to a large part of peninsular Spain and the Balearic Islands. Overall, the spatial distribution of this index was very similar to the mean annual precipitation in Spain, with the highest values in the north-west and in high elevations. Considering events with precipitation over 20 mm (R20mm), the spatial pattern mimicked that of the R10mm.

The uncertainty distribution was very similar for SDII, RX1, and RX5, with higher values along the Mediterranean coast where intense precipitation is more frequent and, consequently, the differences between neighbouring observatories were also higher. R10mm had a low and homogeneous uncertainty all over Spain and R20mm had very low values over central and southern peninsular Spain, in the Balearic Islands and in the eastern Canary Islands.

A clear latitudinal gradient over peninsular Spain was evident for the mean annual maximum consecutive dry days (CDD)

index, with values exceeding 100 days in the south in contrast with less than 20 day in the north (Figure 7 up). The maximum consecutive wet days (CWD) extreme (Figure 7) had a strong longitudinal gradient, with less than 5 days in the east to more than 16 days in the western. The Balearic Islands showed a latitudinal gradient in both indices with more CDD (> 60 days) and less CWD (< 5 days) in the south, coinciding with lower elevations. The Canary Islands had a similar behaviour in all individual islands with the maximum values of CDD (> 110 days) and minimum of CWD (< 5 days).

The 95th percentile of precipitation (R95) showed the maximum values at high elevation areas and in the eastern and southern sides of the Mediterranean coast. This region is considered the central region of peninsular Spain and was more homogeneous with lower values, coinciding with a more continental precipitation regime. The uncertainty values were very low here, which showed the reliability of the estimations of this index. The percentage of precipitation over the 95th percentile contribution to the annual precipitation total (R95rel) showed different patterns, more extreme at eastern peninsular Spain and western Canary Islands. More than the 30% of the precipitation along the Mediterranean coast corresponds to events with amounts of precipitation over the 95th percentile. These values are also common in the Balearic Islands, where all areas had R95rel values over 20%. This spatial distribution represents the extreme character of daily precipitation, especially in Mediterranean areas and in the Ebro Valley.

## 5 Data availability

The SPREAD dataset is freely available in the web repository of the Spanish National Research Council (CSIC). It can be accessed through http://dx.doi.org/10.20350/digitalCSIC/7393, and cited as Serrano-Notivoli et al. (2016). The data is arranged in 6 files (daily precipitation estimations and their uncertainties for peninsular Spain, Balearic Islands and Canary Islands) in NetCDF format that allows an easy processing in scientific analysis software (e.g. R, Python…) and GIS (list of compatible software at http://www.unidata.ucar.edu/software.

## 6 Discussion

High-resolution gridded datasets are useful for regional analysis of daily precipitation, but the accuracy of the estimates depend mainly on the number of available observatories and on the estimation method. Although the method used to build this grid makes independent calculations of each grid point and day, the results showed coherent patterns in spatial distributions of all indices at regional scales.

Some basic parameters of the reconstruction methodology have a key influence in the dissimilarity between different datasets. The selection of one specific method from the many available gridding interpolation methods that can be applied to precipitation may change the final result. (e.g. Creutin and Obled, 1982; Hartkamp et al., 1999; Vicente-Serrano et al., 2003;

Dobesch et al., 2007; Hofstra et al., 2008; Hwang et al., 2012; Brunetti et al., 2014; Militino et al., 2015; Contractor et al., 2015; Herrera et al., 2016). For example, Robeson and Ensor, (2006) and Ensor and Robeson, (2008) argued that the use of geostatistical interpolators for daily precipitation leads to a higher frequency of low-precipitation values while greatly reducing the extreme events. In addition, the high flexibility in the independent variables across the sites allows for a

reasonable estimation of the uncertainty, which is very important for producing datasets that will feed further analyses. Local regressions have been used widely to model daily precipitation with different approaches (Buishand and Tank, 1996; Rajagopalan and Lall, 1998; Marquínez et al., 2003; Simolo et al, 2010; Tardivo and Verti, 2014; Partal et al., 2015), and in this case we used them to compute, from all reconstructed stations, a high-resolution grid estimating separately the probability of a wet/dry day occurrence and the precipitation amount. This two-step procedure avoids an excessive

smoothness of the estimated precipitation fields (Robeson and Ensor, 2005). Furthermore, this individualized calculation allows for an easy update of the dataset, since single days can be reconstructed individually and added to the pre-existing dataset. We also added a measure of uncertainty, which is a big improvement over previous gridded datasets. Uncertainty (which we express by means of the standard error) informs in a quantitative way about the reliability of the estimated data, in a way that can be translated to further calculations such as the daily precipitation indices explored in this article. Our

uncertainty estimation arises from a local interpolation, so it varies spatially and from one day to the next, reflecting the changes in conditions that affect the estimates.

Although some previous datasets exist that obtain daily and/or extreme precipitation indices for Spain, the different methodologies to compute them are a key influence in these differences. The use of the longest precipitation series and the spatial resolution of the final grid, produce very different results. For instance, considering the global datasets, Sillmann et al

(2013) showed for Spain (8 grid points covering the entire peninsular Spain) RX5 values from 50 and 200 mm using the HadEX2 dataset and from 40 and 75 mm using CMIP5 dataset. In the present work, these values ranged between 50 and more than 300 in more than 20,000 grid points. These values were the maximums in HadEX2 and CMIP5 for monsoon areas in southern Asia. Similarly, May (2007), using the HIRHAM model, showed SDII values for Spain between 4.5 and 12.5, while in the present work we found that this index can reach values over 25 mm, especially in the Central Range. Schamm et

al. (2014) used the GPCC (Global Precipitation Climatology Centre) dataset to show values of daily mean precipitation intensity (PMED) between 0.5 and less than 10 mm, while we found a wider range between 3 and more than 15 mm.

Some previous works in Spain that used a lower spatial resolution and a lower number of observatories (Herrera et al., 2012a; Merino et al., 2015) showed overestimated values in extreme precipitation indices in some areas (especially in the

north-west area of the Iberian Peninsula), while smoothing in others (e.g.: Central Range). López-Moreno et al. (2010) showed similar values and spatial patterns in north-east Spain, compared to the ones in the present work for NWD, SDII, CDD and CWD. Despite the relatively high-density station dataset (217 stations) used in their study, they rejected most of the original stations in favour of only the longest series, resulting in a smoothed spatial distribution of the indices, probably also due to the interpolation method (not indicated). Martínez et al. (2007), using 75 stations, showed for Catalonia

(northeast peninsular Spain) the 95th percentile of precipitation values ranging from 20 to 70 mm, which are very similar to those from the SPREAD dataset. All of these works made sub-optimal use of the available data. The values obtained in all of them were correct considering a global conception of precipitation distribution, but daily precipitation requires the highest possible density of observations in order to obtain a proper characterization of its spatial distribution, especially for extreme precipitation. This work provides a representation of the local variability of extremes by using all the available information

and applying a local reconstruction method. If the spatial resolution is amplified for a regional study, based on the use of more precipitation data, the results are less smoothed as shown in Pereira et al. (2016), which used 36 stations in Sierra Nevada (southern peninsular Spain) to compute NWD, R10mm and R20mm with similar values to this work specifying a precise spatial distribution.

## 7 Conclusions

A high-resolution daily precipitation dataset for Spain (SPREAD) is presented. Based on all the available daily precipitation information, a 5 x 5 kilometres spatial resolution grid was built using the *reddPrec* R package (Serrano-Notivoli et al., 2017). The original dataset of observations was quality controlled and the missing values were fitted using the 10 surrounding stations for each day and location to obtain a serially-complete dataset from 1950 to 2012 in peninsular Spain and from 1971 to 2012 in the Balearic and Canary Islands. From this dataset, individual daily precipitation estimations were

computed for each grid point, resulting in a gridded dataset which was consequently used to compute 4 daily precipitation indices and 9 extreme precipitation indices.

PMED showed the highest values in the Central Range and other elevated areas, while NWD, CDDm and CWDm followed a north-west to south-east gradient in peninsular Spain, from high to low values in NWD and CWDm and reverse in CDDm.

5   The south-east of the Iberian Peninsula and the Canary Islands were the driest areas in Spain with less than 30 wet days per year and most than 18 days of the average maximum annual dry spell length. These regions registered less than 2 days of the mean wet event duration.

Extreme precipitation indices showed that the Mediterranean coast is more active in these kind of events, but also that the highest values of SDII, RX1, RX5, R10mm, R20mm and R95 are concentrated in a north-south band of northwest peninsular

10   Spain and, especially, in the Central Range. These results have revealed areas with maximum values not detected in former studies, emphasizing the importance of the use of all available observatories and a sensible methodology that do not produce excessive smoothing while being able to capture local and day-to-day variability.

**Author contribution**

15   M. de Luis and S. Beguería designed the methodological approach in collaboration with R. Serrano-Notivoli, who applied it to the reconstruction of the climate data and the development of the gridded dataset. M.A. Saz and L.A. Longares contributed to the validation process and performed the climatic analysis of the results. R. Serrano-Notivoli prepared the manuscript with contributions from all co-authors.

**Acknowledgements**

This study was supported by research projects CGL2015-69985-R and CGL2014-52135-C3-1-R, financed by the Spanish Ministerio de Economía y Competitividad (MINECO) and FEDER-ERDF funds. The researchers were supported by the Government of Aragón through the 'Programme of research groups' (groups H38, 'Clima, Cambio Global y Sistemas Naturales' and 'E68, Geomorfología y Cambio Global').



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



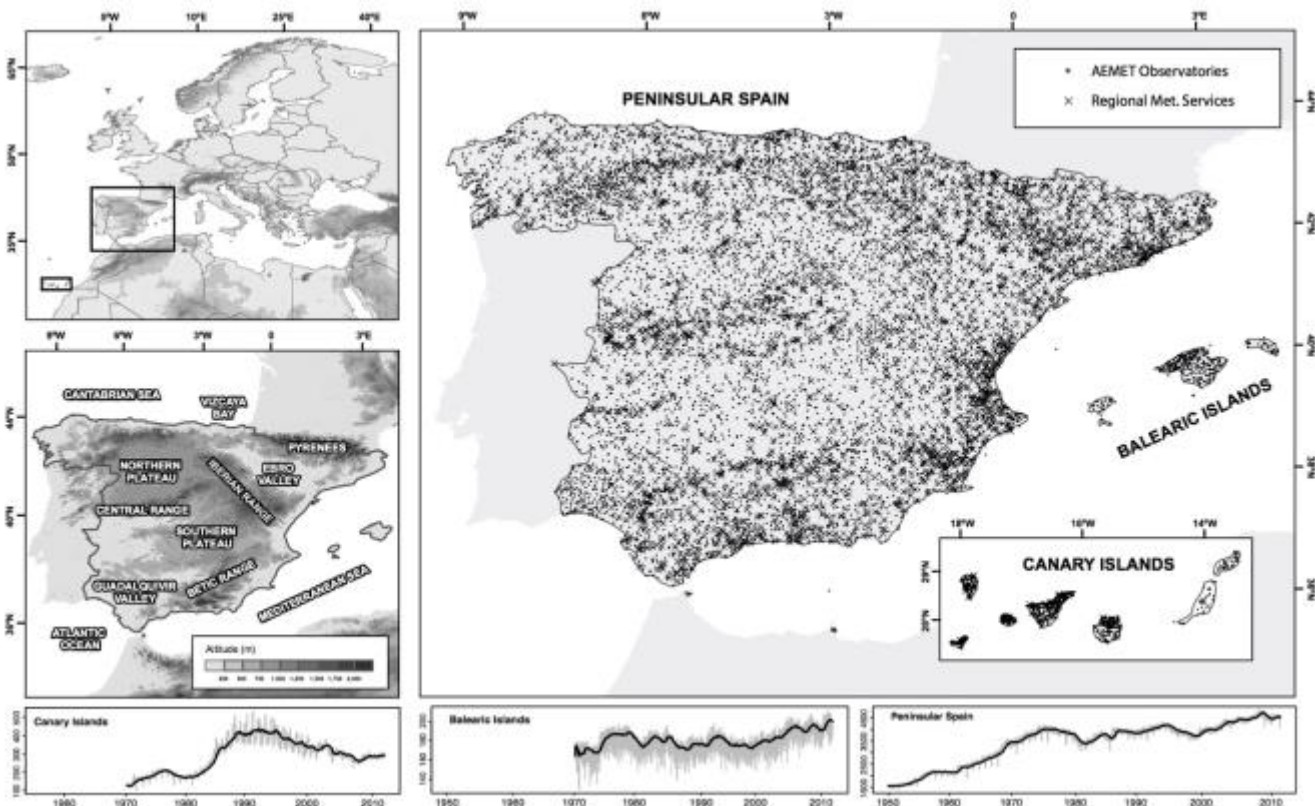

**Figure 1: Location of the precipitation stations used (main map); location of Spain in Europe context (upper left map) and**
5 **geographical references used in the text (lower left map). Number of daily available observatories (grey lines), and its moving average of 365 days (black lines) in Canary Islands (bottom left), Balearic Islands (bottom centre) and Peninsular Spain (bottom right).**





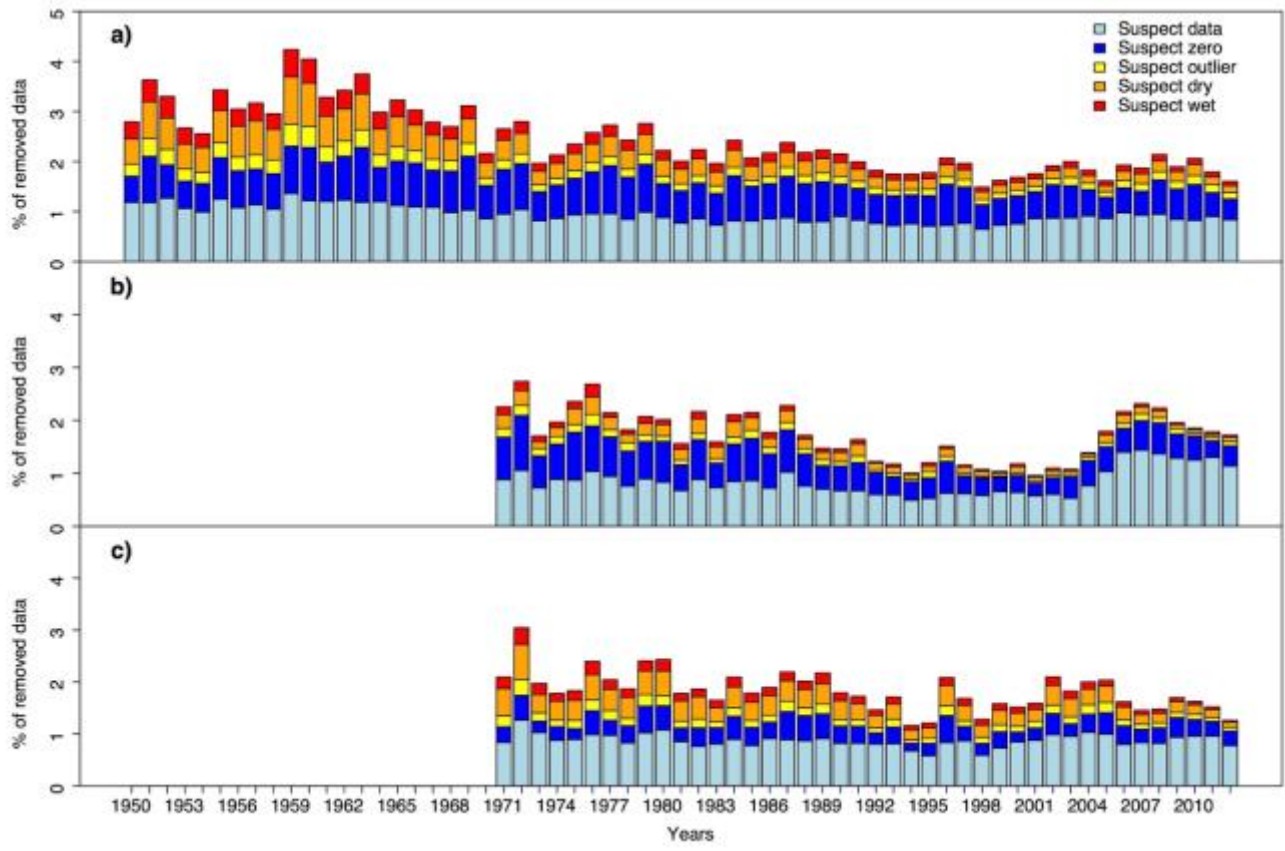

**Figure 2: Removed data by criteria (suspect data, suspect zero, suspect outlier, suspect dry day and suspect wet day): a) Peninsular Spain; b) Balearic Islands and c) Canary Islands.**



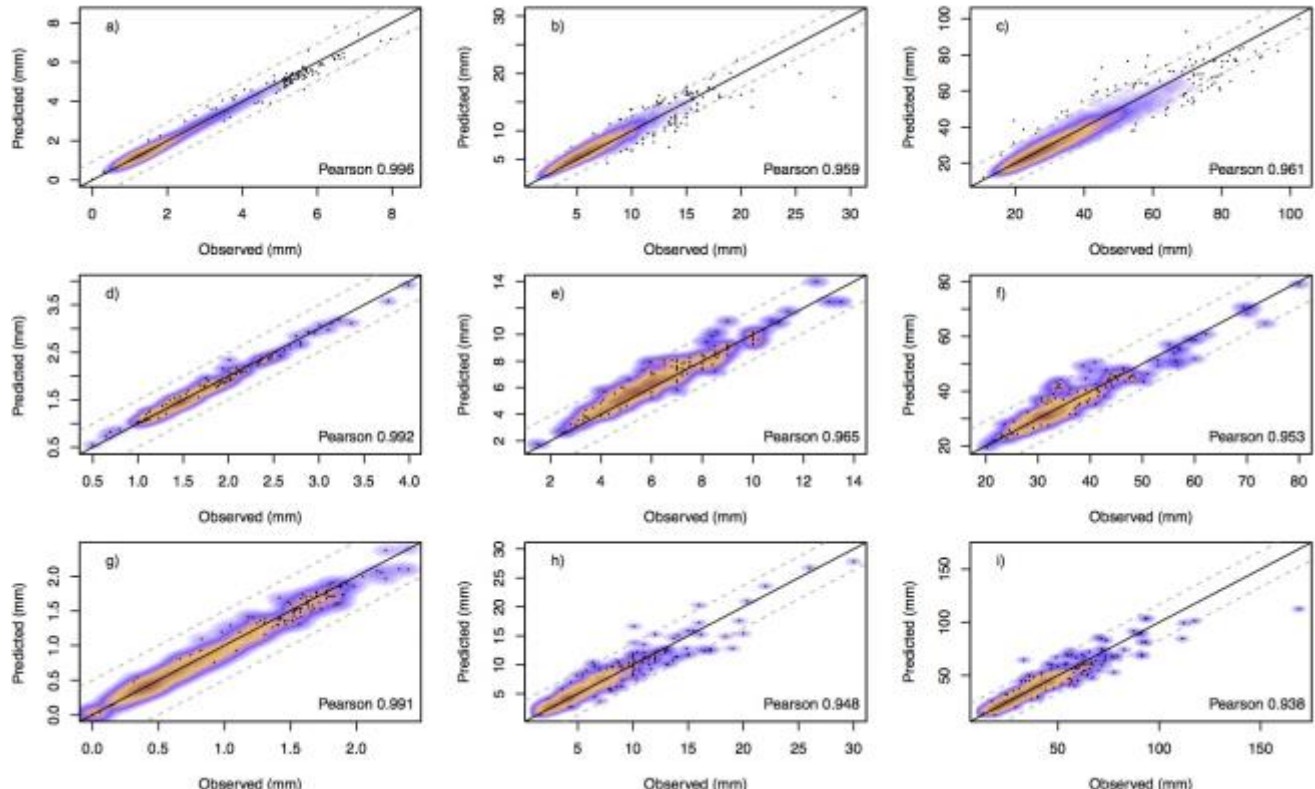

**Figure 3: Scatterplots and Pearson correlation coefficients between observations and estimations of daily precipitation in peninsular Spain (upper line; a, b c), Balearic Islands (mid line: d, e, f) and Canary Islands (bottom line: g, h, i). Dots represent the stations and colours indicate the density. Daily precipitation mean (left column: a, d, g); daily precipitation medians in wet days (central column: b, e, h) and daily precipitation over 95th percentile (right column: c, f, i) are shown.**





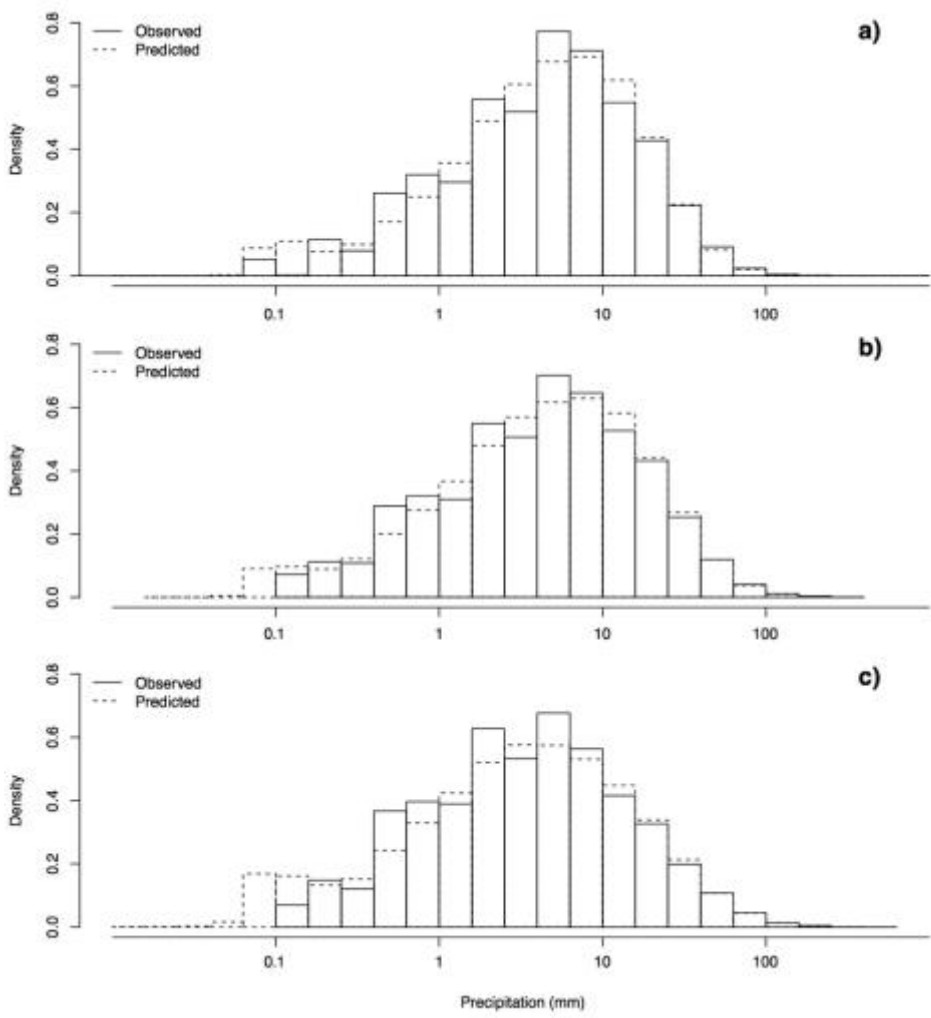

**Figure 4: Histograms of observed and predicted daily precipitation frequency in a) peninsular Spain, b) Balearic Islands and c) Canary Islands.**

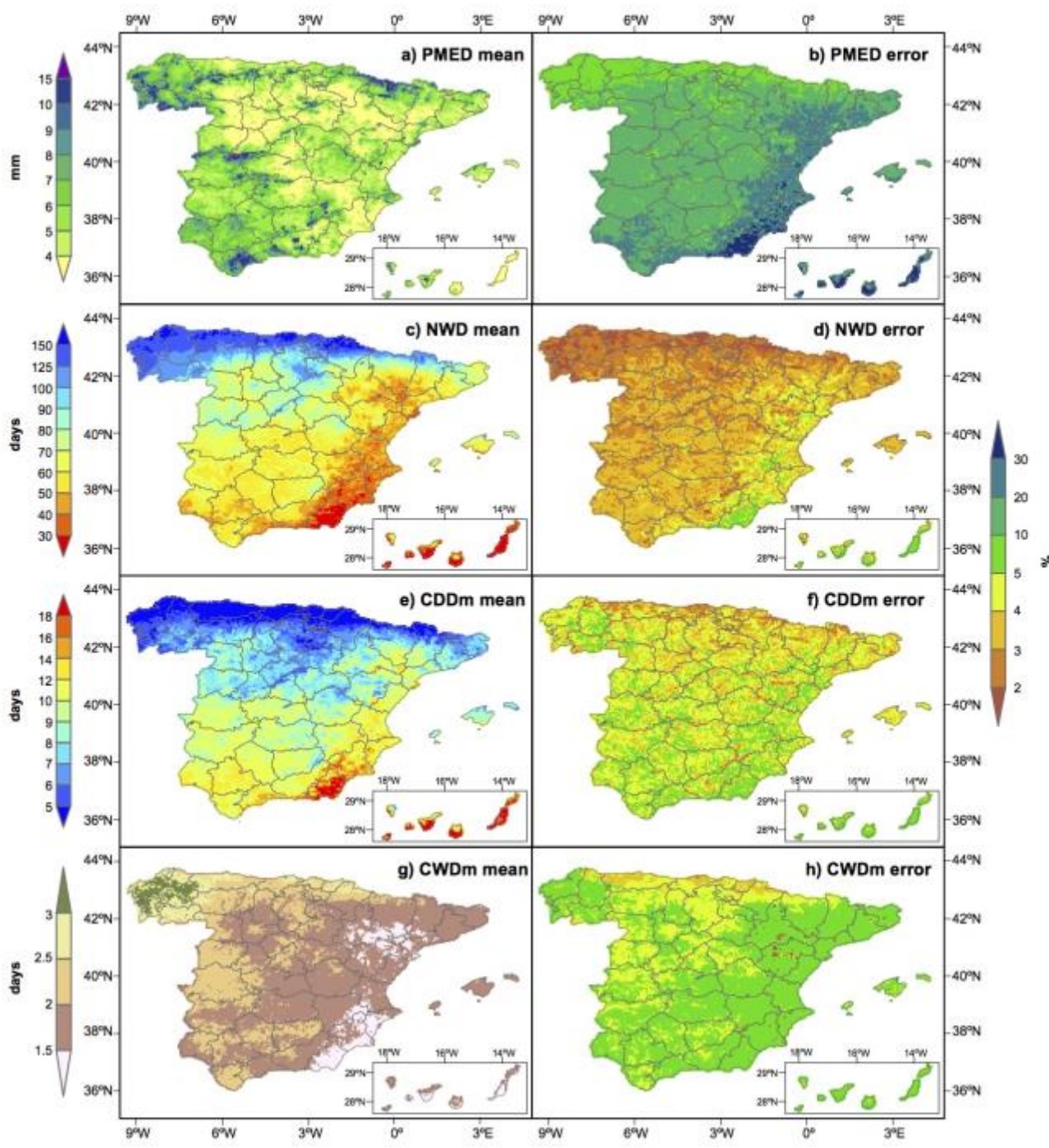

**Figure 5: Daily precipitation indices (left) and their uncertainty (right). PMED: daily mean precipitation intensity; NWD: number of wet days; CDDm: mean consecutive dry days; CWDm: mean consecutive wet days.**







**Figure 6: Daily extreme precipitation indices (left) and their uncertainty (right). SDII: daily precipitation intensity; RX1: maximum 1-day precipitation; RX5: maximum 5-days precipitation; R10mm: Number of days with precipitation over 10 mm; R20mm: Number of days with precipitation over 20 mm.**





**Figure 7: Daily extreme precipitation indices (left) and their uncertainty (right). CDD: maximum consecutive dry days; CWD: maximum consecutive wet days; R95: 95th percentile of precipitation; R95rel: contribution of precipitation over 95th percentile.**



**Table 1: Number of observatories, by source (AEMET: Spanish meteorological agency; MAGRAMA: Ministry of Agriculture and Environment; AHIS: Automatic Hydrological Information System).**

| Source | Number of observatories |
|---|---|
| AEMET | 10,683 (83.1%) |
| MAGRAMA | 541 (4.2%) |
| Meteorological Service of Catalonia | 173 (1.3%) |
| Navarra Government | 38 (0.3%) |
| AHIS of Cantábrico Basin | 36 (0.3%) |
| AHIS of Duero Basin | 216 (1.7%) |
| AHIS of Ebro Basin | 333 (2.6%) |
| AHIS of Guadalquivir Basin | 212 (1.7%) |
| AHIS of Hidrosur Basin | 102 (0.8%) |
| AHIS of Júcar Basin | 182 (1.4%) |
| AHIS of Miño-Sil Basin | 89 (0.7%) |
| AHIS of Segura Basin | 66 (0.5%) |
| AHIS of Tajo Basin | 187 (1.4%) |
| **TOTAL** | **12,858 (100%)** |



**Table 2: Computed indices over daily gridded dataset.**

| Identifier | Description | Units |
|---|---|---|
| *Daily precipitation indices* | | |
| PMED | Daily mean precipitation intensity (*Median of daily precipitation in wet days*) | mm |
| NWD | Number of wet days in a year | Days |
| CDDm | Mean length of dry spell (*Mean number of consecutive dry days*) | Days |
| CWDm | Mean length of wet spell (*Mean number of consecutive wet days*) | Days |
| *Daily extreme precipitation indices* | | |
| SDII | Daily precipitation intensity (*Annual precipitation / number of wet days*) | mm |
| RX1 | Maximum 1-day precipitation | mm |
| RX5 | Maximum 5-day precipitation | mm |
| R10mm | Number of days with precipitation over 10 mm in a year | Days |
| R20mm | Number of days with precipitation over 20 mm in a year | Days |
| CDD | Maximum length of dry spell (*Maximum number of consecutive dry days*) | Days |
| CWD | Maximum length of wet spell (*Maximum number of consecutive wet days*) | Days |
| R95 | 95[th] percentile of daily precipitation in whole series | mm |
| R95rel | Annual contribution of precipitation over 95[th] percentile | % |



**Table 3: Accuracy of the wet/dry days estimates: percent observed dry (P=0) and wet (P>0) days, and percent predicted dry (RV=0) and wet (RV>0) days on observed dry and wet days.**

|  | Peninsular Spain | | Balearic Islands | | Canary Islands | |
| --- | --- | --- | --- | --- | --- | --- |
|  | **P=0** | **P>0** | **P=0** | **P>0** | **P=0** | **P>0** |
| **Observed** | 79.26 | 20.74 | 81.59 | 18.41 | 89.96 | 10.04 |
| **RV=0** | 94.84 | 20.47 | 96.35 | 18.11 | 96.76 | 29.46 |
| **RV>0** | 5.16 | 79.53 | 3.65 | 81.89 | 3.24 | 70.54 |



**Table 4: The leave-one-out cross-validation (LOO-CV) statistics showing the goodness of fit between observations and estimations of daily precipitation separated by altitudes (m. a.s.l.). IP: Iberian Peninsula; BI: Balearic Islands; CI: Canary Islands; N: number of stations; MAE: Mean Absolute Error; ME: Mean Error; %OBS: Percentage of observed precipitation; %PRE: Percentage of predicted precipitation; RM: Ratio of Means; RSD: Ratio of Standard Deviations. Results were constrained to 2 decimal places.**

| | | 0-100 | >100-300 | >300-500 | >500-700 | >700-900 | >900-1,100 | >1,100-1,300 | >1,300-1,500 | > 1,500-2,000 | >2,000 |
|---|---|---|---|---|---|---|---|---|---|---|---|
| | N | 918 | 1,054 | 1,077 | 1,210 | 1,271 | 766 | 415 | 128 | 65 | 11 |
| | %OBS | 12.50 | 14.40 | 14.20 | 17.20 | 17.30 | 12.70 | 7.90 | 2.30 | 1.30 | 0.20 |
| | %PRE | 12.60 | 14.40 | 14.20 | 17.20 | 17.30 | 12.70 | 7.80 | 2.30 | 1.40 | 0.20 |
| IP | RM | 1.00 | 0.99 | 0.99 | 0.99 | 0.99 | 0.99 | 0.98 | 1.01 | 1.06 | 1.07 |
| | RSD | 0.95 | 0.96 | 0.96 | 0.96 | 0.97 | 0.96 | 0.96 | 1.03 | 1.10 | 1.19 |
| | ME | 0.04 | -0.02 | -0.02 | -0.01 | 0.01 | 0.00 | -0.03 | 0.26 | 0.64 | 1.47 |
| | MAE | 4.83 | 4.55 | 4.55 | 4.51 | 4.3 | 4.86 | 5.48 | 6.32 | 6.92 | 9.61 |
| | N | 128 | 93 | 20 | 4 | 2 | 1 | 0 | 0 | 0 | 0 |
| | %OBS | 43.80 | 39.20 | 12.80 | 2.70 | 1.10 | 0.40 | | | | |
| | %PRE | 44.00 | 39.00 | 12.90 | 2.60 | 1.10 | 0.40 | | | | |
| BI | RM | 1.02 | 0.99 | 1.03 | 0.84 | 1.09 | 1.43 | | | | |
| | RSD | 1.00 | 0.96 | 0.97 | 0.96 | 0.99 | 1.34 | | | | |
| | ME | 0.17 | -0.10 | 0.11 | -0.48 | 0.52 | 2.65 | | | | |
| | MAE | 3.78 | 3.82 | 5.71 | 7.76 | 10.11 | 8.12 | | | | |
| | N | 78 | 96 | 79 | 70 | 40 | 37 | 13 | 9 | 9 | 10 |
| | %OBS | 8.40 | 15.00 | 18.70 | 20.50 | 13.40 | 11.00 | 5.20 | 4.30 | 2.20 | 1.30 |
| | %PRE | 8.80 | 15.20 | 18.40 | 20.40 | 13.20 | 10.90 | 5.20 | 4.20 | 2.30 | 1.40 |
| CI | RM | 1.02 | 1.02 | 0.97 | 1.00 | 1.02 | 0.99 | 0.98 | 1.02 | 1.01 | 0.92 |
| | RSD | 1.02 | 1.02 | 0.98 | 0.99 | 0.98 | 0.99 | 1.05 | 0.96 | 1.03 | 0.93 |
| | ME | 0.39 | 0.17 | -0.15 | 0.00 | -0.13 | -0.15 | -0.02 | -0.16 | 0.48 | 1.85 |
| | MAE | 4.49 | 4.18 | 5.28 | 5.91 | 5.94 | 5.95 | 6.26 | 5.09 | 5.48 | 14.28 |



**Table 5: The leave-one-out, cross-validation (LOO-CV) statistics, showing the goodness of fit between observations and estimates of monthly aggregates. IP: Iberian Peninsula; BI: Balearic Islands; CI: Canary Islands; MAE: mean absolute error; ME: mean error; RM: ratio of means; RSD: ratio of standard deviations. Results were constrained to 2 decimal places.**

|     |      | JAN   | FEB  | MAR  | APR   | MAY   | JUN  | JUL   | AUG   | SEP   | OCT   | NOV   | DEC   |
|-----|------|-------|------|------|-------|-------|------|-------|-------|-------|-------|-------|-------|
| IP  | MAE  | 10.18 | 9.66 | 9.28 | 10.88 | 11.05 | 8.39 | 4.49  | 6.00  | 9.68  | 12.10 | 13.47 | 13.59 |
|     | ME   | 0.45  | 0.40 | 0.44 | 0.60  | 0.53  | 0.13 | -0.07 | -0.01 | 0.32  | 0.66  | -4.04 | -4.30 |
|     | RM   | 1.01  | 1.01 | 1.01 | 1.01  | 1.01  | 1.00 | 0.99  | 1.00  | 1.01  | 1.01  | 0.94  | 0.93  |
|     | RSD  | 1.02  | 1.02 | 1.03 | 1.03  | 1.03  | 1.03 | 1.02  | 1.03  | 1.03  | 1.02  | 0.94  | 0.94  |
| BI  | MAE  | 8.82  | 7.40 | 7.35 | 8.10  | 7.32  | 4.27 | 2.72  | 6.39  | 13.25 | 14.32 | 14.73 | 13.18 |
|     | ME   | 0.42  | 0.31 | 0.49 | 0.38  | 0.24  | 0.15 | 0.04  | 0.01  | 0.40  | 1.07  | -3.72 | -3.30 |
|     | RM   | 1.01  | 1.01 | 1.01 | 1.01  | 1.01  | 1.01 | 1.01  | 1.00  | 1.01  | 1.01  | 0.96  | 0.95  |
|     | RSD  | 1.02  | 1.02 | 1.01 | 1.01  | 1.01  | 1.02 | 1.02  | 1.02  | 1.01  | 1.03  | 0.96  | 0.96  |
| CI  | MAE  | 8.91  | 8.61 | 8.33 | 5.44  | 2.52  | 1.07 | 0.52  | 0.79  | 2.89  | 6.85  | 12.10 | 13.95 |
|     | ME   | 0.00  | 0.10 | 0.01 | 0.00  | -0.08 | 0.00 | 0.00  | -0.03 | -0.10 | -0.01 | -4.5  | -4.18 |
|     | RM   | 1.00  | 1.00 | 1.00 | 1.00  | 0.99  | 1.00 | 1.01  | 0.98  | 0.99  | 1.00  | 0.90  | 0.93  |
|     | RSD  | 1.01  | 1.02 | 1.06 | 1.06  | 1.08  | 1.11 | 1.19  | 1.10  | 1.07  | 1.05  | 0.93  | 0.91  |



**Table 6: The leave-one-out, cross-validation (LOO-CV) statistics, showing the goodness of fit between observations and estimates of daily and extreme precipitation indices. IP: Iberian Peninsula; BI: Balearic Islands; CI: Canary Islands; MAE: mean absolute error; ME: mean error; RM: ratio of means; RSD: ratio of standard deviations. Results were constrained to 2 decimal places.**

| | | RX1 | RX5 | R10mm | R20mm | CDD | CWD | SDII | P95 | R95rel | NWD | PMED | CDDm | CWDm |
|---|---|---|---|---|---|---|---|---|---|---|---|---|---|---|
| **IP** | MAE | -2.42 | 0.15 | -0.01 | -0.02 | 0.02 | -0.01 | -0.31 | 0.02 | 0.00 | -0.02 | -0.09 | 0.00 | 0.00 |
| | ME | 4.64 | 2.49 | 0.05 | 0.03 | 0.37 | 0.07 | 1.27 | 0.57 | 0.25 | 0.38 | 1.29 | 0.08 | 0.01 |
| | RM | 0.96 | 1.01 | 0.99 | 0.95 | 1.02 | 1.01 | 1.00 | 1.00 | 1.00 | 1.02 | 1.09 | 1.01 | 1.00 |
| | RSD | 0.98 | 1.01 | 1.00 | 0.97 | 1.03 | 1.01 | 1.04 | 0.97 | 1.08 | 1.01 | 1.14 | 1.03 | 1.02 |
| **BI** | MAE | -2.18 | -0.35 | -0.01 | -0.01 | -0.06 | -0.01 | -0.22 | 0.14 | 0.02 | -0.07 | -0.12 | 0.01 | 0.00 |
| | ME | 5.25 | 3.89 | 0.05 | 0.03 | 0.47 | 0.06 | 1.35 | 0.45 | 0.2 | 0.52 | 1.33 | 0.09 | 0.01 |
| | RM | 0.97 | 1.00 | 1.00 | 0.99 | 1.00 | 1.00 | 1.00 | 1.01 | 1.00 | 1.01 | 1.06 | 1.02 | 1.00 |
| | RSD | 0.99 | 1.01 | 1.02 | 1.01 | 1.02 | 1.01 | 1.00 | 1.02 | 1.03 | 1.02 | 1.09 | 1.24 | 1.01 |
| **CI** | MAE | -1.22 | 0.47 | -0.01 | 0.00 | -0.22 | -0.01 | -1.23 | -0.1 | 0.10 | -0.05 | -1.17 | -0.07 | 0.00 |
| | ME | 4.80 | 3.79 | 0.04 | 0.02 | 0.86 | 0.06 | 2.40 | 0.48 | 0.30 | 0.37 | 2.20 | 0.27 | 0.01 |
| | RM | 1.00 | 1.04 | 1.01 | 1.00 | 0.99 | 1.01 | 0.95 | 0.97 | 1.01 | 1.13 | 0.98 | 0.99 | 0.99 |
| | RSD | 1.01 | 1.05 | 1.01 | 1.02 | 1.01 | 0.98 | 1.01 | 0.99 | 1.40 | 1.07 | 1.19 | 1.16 | 0.97 |

