# Peer review of "SPREAD: A high-resolution daily gridded precipitation dataset for Spain"

_Earth System Science Data, 2017_

## Referee Comment (RC1) · Anonymous Referee #1 · 3 Jul 2017

**REVISION (essd-2017-35)**

**SPREAD: A high-resolution daily gridded precipitation dataset for Spain**

**GENERAL COMMENTS**

The manuscript is very interesting dealing about a public high-resolution daily gridded precipitation dataset for Spain and Balearic and Canary Islands. This dataset is very important and useful because data can be used in other type of studies. On the other hand, authors used the dataset to analyze the frequency and intensity of extreme precipitation events and the spatial distribution of daily precipitation in peninsular Spain, Baleares and Canarias.

**SPECIFIC COMMENTS**

- I suggest to modify the title to authors because apart from building SPREAD they analyze the frequency and intensity of extreme precipitation events and the spatial distribution of daily precipitation using different indices. Some conclusions of these analyses should be also included in the abstract.

- Pg. 2: Martínez et al, 2007 is not in the references list

- Pg. 6: Keggenhof et al, 2014 is not in the references list

- Pg 13: Herrero et al, 2016; Militino et al, 2015; Marquínez et al, 2003 are not in the references list

- Pg 14: Herrera et al, 2012a is not in the references list

- Martínez et al, 2010 is not in the main text

- Please, arrange references according to the guidelines.

---

## Referee Comment (RC2) · Anonymous Referee #2 · 24 Jul 2017

General comments: This manuscript is devoted to the development of a new high-resolution daily gridded precipitation dataset for Spain using a large number of stations (12 858) and over the period from 1950 to 2012 (peninsular Spain) or from 1971 to 2012 (Balearic and Canary Archipelagos). Data is available over a 5 km grid. The dataset is publically available to users and the authors provide not only precipitation estimates, but also their corresponding uncertainties. The dataset is properly validated with observational data. The methodology is adequate and based on previous studies from the authors, namely on their R-package 'reddPrec'. Furthermore, a number of precipitation indices is also analysed, including indices of extremes. The text is generally clear and well written. The high-resolution dataset produced by this study is of

major relevance for impact assessments over a wide range of socioeconomic sectors and for decision-making. Therefore, I recommend the publication of this manuscript after some very minor revisions detailed below.

Specific comments: 1. The title should explicitly mention that a climatological analysis is undertaken, including an analysis of precipitation extremes.

2. Section 2: I would like to see here more discussion regarding the implications of the data gaps on the results. As the authors mention, only 17 stations actually cover the full period. Although the station density remains reasonably high throughout the whole period (please revise Y-axis labels in the bottom panels of Fig. 1), some important limitations/uncertainties are expected to arise from this lack of data. Please enhance this discussion.

3. Page 5, Lines 9-11: the definitions of suspect wet and dry days seem to be reversed.

4. Section 3: from my viewpoint, this section should provide further details concerning the followed methodology. I understand that there are limitations in the paper extent, but a deeper description of the methods should improve the readability of the text and may prevent readers from reading preceding papers.

———————————————————

---

## Referee Comment (RC3) · Anonymous Referee #3 · 25 Jul 2017

**GENERAL COMMENTS**

In my opinion, this paper is timely and necessary. It is a research developed with rigor, very well documented. The writing is clear and simple. The methodology is applicable in any other territory. Undoubtedly, this research will be useful for policy makers and managers or administrators of the territory. This paper can become a basic reference for those researchers interested in the evolution of rainfall and the changes that are happening in Spain, a country that can be seriously affected, and in a few years, by the so-called climate change. Tables and figures are appropriate. All these reasons justify a positive, very favourable overall assessment: excellent.

[Figure]

SPECIFIC COMMENTS

- I suggest, like other reviewers, to complete the title of the paper. Add "Potential applications" ??.

- Review citations and final references. Some citations are not referenced and some references do not appear in the main text (for example: Martínez et al, 2010).

- Organize the references according to the rules for authors.

I am grateful for the opportunity to review this paper, as it will be a good support for my future research.

---

## Referee Comment (RC4) · Anonymous Referee #4 · 4 Aug 2017

General comments:

This paper presents a new high resolution gridded daily precipitation product for Spain (Peninsula and Isles) using a state-of-the-art method. The dataset is of high relevance for many applications. The fact that the grid of uncertainties is provided as part of the dataset adds a lot of value to it. The article is well written, clear and easy to follow. The example of application of the dataset (indices of extremes) presented in the article is interesting and adds scientific value to the paper. The figures and tables are all pertinent and well described. The methods used to derive the dataset are not new. However, to my knowledge, no other such high resolution daily precipitation gridded product is

publicly available in Spain. This dataset will be hugely valuable to a range of users. It will contribute to avoid an enormous amount of future duplicated effort by making available a fully validated high quality product for precipitation in Spain. Until now, each individual research group would produce their own gridded data, generally with lower quality due to lack of time or skills. Therefore this dataset will contribute to improve the quality of future research in a wide variety of fields as precipitation is relevant to so many disciplines (catchment hydrology, drought/floods studies, groundwater recharge, nitrogen deposition, ecosystems, agriculture, land surface processes, epidemiology, amongst many others). The fact that the values of uncertainty are included informs the user on the quality of the data which allows to make an informed decision on whether to trust the data at a given time and location. Uncertainty values are also a requisite for applying data assimilation techniques. The dataset will be relevant not only for scientists but also to many stakeholders (water managers, policy-makers, NGOs). For all the above, this dataset could potentially become the new reference high quality dataset for precipitation for Spain. Therefore, I would recommend the publication of this paper after some minor revisions described below.

Specific comments:

I. The manuscript:

The manuscript is generally well written, easy to follow and well structured. Below are a few specific comments to help improve some minor aspects of the paper.

1) In the introduction, page 1, line 27, the authors say "High-resolution spatiotemporal precipitation datasets are useful tools for land management", and that is the only example of potential application of their dataset they give. I can think of many more disciplines which would benefit from the existence of such a dataset, and I feel the introduction is not doing justice to the importance of this new dataset. The authors should emphasise much more on the value and impact this dataset will have in the scientific (and non-scientific) community for future studies in Spain. In the general comments

above, I give a few examples of fields where such gridded meteorological datasets are in high demand. The authors could use that list as a starting point to expand their introduction. A reader should be more convinced on why this dataset is so important, the introduction should include some examples of applications with a few key publications illustrating that (for example from other countries where such dataset already exist). This dataset is really of high value, so this should be reflected in the introduction.

2) Page 2, line 14: Brunet et al., 2006. This paper was about gridded temperature dataset, so probably not too relevant here, unless you add the word "meteorological" before "datasets" in the sentence "a few daily datasets have been made for Spain or some of its regions".

3) Page 2, line 24-25: "The uncertainties of these estimations depend on the density of observations used to compute the model": you don't mention once in your paper the uncertainties coming from the uncertainties in the raw raingauge data. See comment below number 5.

4) Page 3, line 18-20: Please include the areas in km2 of Peninsular Spain, Balearic Islands and Canary Islands so that a non-Spanish reader gets a quick feel of the density of each network.

5) Input data (page 3-4): Are the instruments uniform across the network? Which type of raingauge are they (automated or manual, or a mixture)? You do some quality control checks on your input data, which is very valuable. However, raingauge measurements are notoriously uncertain, especially in certain circumstances such as windy conditions (Rodda and Dixon, 2012), leading to significant undercatches, up to 16% in highly exposed areas (Rodda and Smith, 1986). Whereas these uncertainties are very difficult to estimate in practice, they should at least be acknowledged somewhere in your paper.

Rodda, J. C. and Dixon, H.: Rainfall measurement revisited, Weather, 67, 131-136, 2012.

Rodda, J. C. and Smith, S. W.: The significance of the systematic error in rainfall measurement for assessing wet deposition, Atmospheric Environment (1967), 20, 1059-1064, 1986.

6) Input data (page 3-4): When you measure precipitation, I assume this includes both rainfall and the water equivalent of new snow. It is probably worth stating that explicitly in the text.

7) Page 8 line 12: "greater radius": how much greater? If possible, it would be interesting to know the distance to the closest and the furthest 10 neighbours (maximum and mean) in Peninsular, Canary and Balearic Isles.

8) Page 9, line 10: "These months": which months are you referring to? November and December? Not clear from the text.

9) Page 10: Analysis on NWD, CDDm and CWDm: very interesting. Although it would have been even more interesting if this was done per year or per decade which would have allowed to see the evolution over 60 years. It would be very nice if this could be included in the application example, but the authors might feel this goes beyond the scope of this paper.

10) Page 12, line 16-17: see comments 3 and 5.

11) Page 32, Table 5: How come MAE for the Peninsula has systematically higher values than in table 4?

12) Page 33, Table 6: Please re-order the indices so that they are in the same order as in table 2. This would improve the readability greatly.

II. The dataset:

The dataset was very easily downloadable, with adequate metadata. Ideally, it would have been good if the data complied with some existing international standards or conventions for NetCDF files, such as CF conventions. These conventions are developed

to promote the interchange and sharing of netCDF files in a common format. Some tools or software require netcdf files to be CF or COARDS compliant. One recommendation of the CF conventions (which is compulsory for COARDS conventions) is that, for a multidimensional variable (such as the precipitation in this dataset), the order of the dimensions should be time, (height if present), lat (or y), lon (or x). This is not the case for this dataset, in which x comes before y, which means it can't be used with some tools. It is good practice to try and comply as much as possible to existing international standards so that datasets are as usable to as many users as possible. Having said that, this doesn't reduce the value of the dataset itself, which is still easily usable, and the order of the dimensions could be reasonably quickly fixed by the user if needed be.

Technical corrections:

Spelling: the use of English is inconsistent, the authors sometimes use British spelling (kilometre, neighbour, neighbouring, behaviour, favour, analyse) and sometimes American spelling (characterize, generalized, individualized, characterization, emphasizing). The authors should chose either British or American English and correct the spelling accordingly.

Page 3 line 11: remove the word "make"

Page 15 line 6: replace "most" by "more"

---

## Author Comment (AC1) · 11 Aug 2017

**RESPONSE LETTER**

**Review of Manuscript No.: essd-2017-35**
**Title: SPREAD: A high-resolution daily gridded precipitation dataset for Spain**
**Author(s):** Serrano-Notivoli R., Beguería S., Saz M.A., Longares, L.A. and De Luis M.

**POINT BY POINT: REVIEWER #1 COMMENTS:**

**GENERAL COMMENTS**
**The manuscript is very interesting dealing about a public high-resolution daily gridded precipitation dataset for Spain and Balearic and Canary Islands. This dataset is very important and useful because data can be used in other type of studies. On the other hand, authors used the dataset to analyze the frequency and intensity of extreme precipitation events and the spatial distribution of daily precipitation in peninsular Spain, Baleares and Canarias.**

> Thank you for your comments. We also think that this dataset could be useful to many other disciplines. The public availability of the data will help the researchers to deal with a final and reliable product of climate data.

**SPECIFIC COMMENTS**
**- I suggest to modify the title to authors because apart from building SPREAD they analyze the frequency and intensity of extreme precipitation events and the spatial distribution of daily precipitation using different indices. Some conclusions of these analyses should be also included in the abstract.**

> A short statement in the title has been added as suggested to indicate the analysis of the extreme precipitation developed in the manuscript. The title is now: "*SPREAD: A high-resolution daily gridded precipitation dataset for Spain. An extreme events frequency and intensity overview.*".

> Also, main conclusions of the extreme precipitation analysis have been added in the abstract (see page 1; lines 17-19).

**- Pg. 2: Martínez et al, 2007 is not in the references list**

> Sorry for mistake. The year of publication was wrong, the correct reference is in the list now.

**- Pg. 6: Keggenhof et al, 2014 is not in the references list**

> Thanks; the reference has been added to the list.

**- Pg 13: Herrero et al, 2016; Militino et al, 2015; Marquínez et al, 2003 are not in the references list**

> Thanks; the references have been added to the list.

**- Pg 14: Herrera et al, 2012a is not in the references list**

> Sorry for mistake. The correct reference is "Herrera et al., 2012". Corrected in the text.

**- Martínez et al, 2010 is not in the main text**

> Thanks; this reference was wrong. It has been removed from the references list.

**- Please, arrange references according to the guidelines.**

All the references have been revised and changed according to the guidelines of the journal.

---

## Author Comment (AC2) · 11 Aug 2017

**RESPONSE LETTER**

**Review of Manuscript No.: essd-2017-35**
**Title : SPREAD: A high-resolution daily gridded precipitation dataset for Spain**
**Author(s):** Serrano-Notivoli R., Beguería S., Saz M.A., Longares, L.A. and De Luis M.

**POINT BY POINT: REVIEWER #2 COMMENTS:**

**GENERAL COMMENTS**
**This manuscript is devoted to the development of a new high- resolution daily gridded precipitation dataset for Spain using a large number of stations (12 858) and over the period from 1950 to 2012 (peninsular Spain) or from 1971 to 2012 (Balearic and Canary Archipelagos). Data is available over a 5 km grid. The dataset is publically available to users and the authors provide not only precipitation estimates, but also their corresponding uncertainties. The dataset is properly validated with observational data. The methodology is adequate and based on previous studies from the authors, namely on their R-package 'reddPrec'. Furthermore, a number of precipitation indices is also analysed, including indices of extremes. The text is generally clear and well written. The high-resolution dataset produced by this study is of major relevance for impact assessments over a wide range of socioeconomic sectors and for decision-making. Therefore, I recommend the publication of this manuscript after some very minor revisions detailed below.**

> Thank you for your kind and positive comments. Indeed, the SPREAD dataset is not only a final product in itself, it could help in a wide range of decision-making policies.

**SPECIFIC COMMENTS**
**1. The title should explicitly mention that a climatological analysis is undertaken, including an analysis of precipitation extremes.**

> As indicated in previous comments in the response to reviewer#1, a short statement in the title has been added as suggested to indicate the analysis of the extreme precipitation developed in the manuscript. The title is now: "*SPREAD: A high-resolution daily gridded precipitation dataset for Spain. An extreme events frequency and intensity overview.*".

**2. Section 2: I would like to see here more discussion regarding the implications of the data gaps on the results. As the authors mention, only 17 stations actually cover the full period. Although the station density remains reasonably high throughout the whole period (please revise Y-axis labels in the bottom panels of Fig. 1), some important limitations/uncertainties are expected to arise from this lack of data. Please enhance this discussion.**

> Indeed, only a few stations cover the complete period. However, the use of the complete stations network helped to a more reliable daily estimate of precipitation in longest data series. Only reconstructed stations with more than 10 years of original data were used to build the grid. The use of short series introduces more missing data in the whole dataset, but estimating precipitation day by day with the 10 nearest neighbours there are more probabilities of finding near data, which improves the final estimate.
>
> A new paragraph has been added at the end of the discussion section:
> *"The use of the complete information of the precipitation network in Spain provided a more detailed precipitation distribution over time and space. Although only a few stations covered the complete period, the use of short data series helped to estimate the missing precipitation values in longer ones, which were used to build the whole grid. A high number of grid points (higher spatial resolution) in combination with a low-density*

*stations network could lead into higher uncertainties. This work aimed to set a compromise between both factors by using a high number of stations and a mid-high spatial resolution. In addition, the magnitude of the uncertainty informed about the reliability of each estimate. A higher uncertainty means more differences between the data used to estimate precipitation and these differences can be increased with a lower number of stations."*

**3. Page 5, Lines 9-11: the definitions of suspect wet and dry days seem to be reversed.**

Right, "wet" and "dry" words were switched. Sorry for mistake.

**4. Section 3: from my viewpoint, this section should provide further details concerning the followed methodology. I understand that there are limitations in the paper extent, but a deeper description of the methods should improve the readability of the text and may prevent readers from reading preceding papers.**

As you indicate in your comment, there are limitations in the paper extent. The basics of the methodology were tried to be explained in the text: The reference values (RV) used for quality control and reconstruction (3.1) and the gridding (3.2). Both the reconstruction of the original series and the new estimates for the grid points are based on the computation of the RV, which are calculated with GLM (Generalized Linear Models) based on the 10 nearest neighbors. We tried to synthetize the whole method in section 3, and we thought that, as this work is not only methodological but used to present and validate a data product, the methods basics were enough to the understanding of the grid creation process. The cited previous work, Serrano-Notivoli et al., 2017 (now, 2017a), was referred to the R package used to build the grid. However, we added a new reference (Serrano-Notivoli et al., 2017b) that widely explains the details of the method that we think are not essential for this product presentation.

Serrano-Notivoli, R., de Luis, M., Saz, M.A., Beguería, S.: Spatially-based reconstruction of daily precipitation instrumental data series, Clim. Res., doi: 10.3354/cr01476, In press, 2017b.

We made an effort to summarize the methodological part because, as it is already published, we tried to avoid duplicities. Anyway, we can extend the explanation of any specific point if requested. Please, let us know and we will be pleased to do that.

---

## Author Comment (AC3) · 11 Aug 2017

**RESPONSE LETTER**

**Review of Manuscript No.: essd-2017-35**
**Title: SPREAD: A high-resolution daily gridded precipitation dataset for Spain**
**Author(s):** Serrano-Notivoli R., Beguería S., Saz M.A., Longares, L.A. and De Luis M.

**POINT BY POINT: REVIEWER #3 COMMENTS:**

**GENERAL COMMENTS**
**In my opinion, this paper is timely and necessary. It is a research developed with rigor, very well documented. The writing is clear and simple. The methodology is applicable in any other territory. Undoubtedly, this research will be useful for policy makers and managers or administrators of the territory. This paper can become a basic reference for those researchers interested in the evolution of rainfall and the changes that are happening in Spain, a country that can be seriously affected, and in a few years, by the so-called climate change. Tables and figures are appropriate. All these reasons justify a positive, very favourable overall assessment: excellent.**

> Thank you for your kind comments. As stated in previous comments to the rest of the reviewers, our purpose with this work is to serve as basis to further researches and decision-making policies. Certainly, the precipitation in Spain has changed in last decades and currently the authors are working in the study of recent evolution of this climate parameter.

**SPECIFIC COMMENTS**
**- I suggest, like other reviewers, to complete the title of the paper. Add "Potential applications" ??.**

> Thank you for the suggestion. As many of the reviewers have focused in this point, a short statement in the title has been added to indicate the analysis of the extreme precipitation developed in the manuscript. The title is now: "*SPREAD: A high-resolution daily gridded precipitation dataset for Spain. An extreme events frequency and intensity overview.*".

**- Review citations and final references. Some citations are not referenced and some references do not appear in the main text (for example: Martínez et al, 2010).**
**- Organize the references according to the rules for authors.**

> Sorry for that. Some of the references were wrongly cited or referenced. We have reviewed all of them.

**I am grateful for the opportunity to review this paper, as it will be a good support for my future research.**

> Thanks, we are grateful to you and the rest of the reviewers for investing your time and effort to read the manuscript and make useful suggestions.

---

## Author Comment (AC4) · 11 Aug 2017

**RESPONSE LETTER**

Review of Manuscript No.: essd-2017-35
Title: SPREAD: A high-resolution daily gridded precipitation dataset for Spain
Author(s): Serrano-Notivoli R., Beguería S., Saz M.A., Longares, L.A. and De Luis M.

**POINT BY POINT: REVIEWER #4 COMMENTS:**

**GENERAL COMMENTS**
This paper presents a new high resolution gridded daily precipitation product for Spain (Peninsula and Isles) using a state-of-the-art method. The dataset is of high relevance for many applications. The fact that the grid of uncertainties is provided as part of the dataset adds a lot of value to it. The article is well written, clear and easy to follow. The example of application of the dataset (indices of extremes) presented in the article is interesting and adds scientific value to the paper. The figures and tables are all pertinent and well described. The methods used to derive the dataset are not new. However, to my knowledge, no other such high resolution daily precipitation gridded product is publicly available in Spain. This dataset will be hugely valuable to a range of users. It will contribute to avoid an enormous amount of future duplicated effort by making available a fully validated high quality product for precipitation in Spain. Until now, each individual research group would produce their own gridded data, generally with lower quality due to lack of time or skills. Therefore this dataset will contribute to improve the quality of future research in a wide variety of fields as precipitation is relevant to so many disciplines (catchment hydrology, drought/floods studies, groundwater recharge, nitrogen deposition, ecosystems, agriculture, land surface processes, epidemiology, amongst many others). The fact that the values of uncertainty are included informs the user on the quality of the data which allows to make an informed decision on whether to trust the data at a given time and location. Uncertainty values are also a requisite for applying data assimilation techniques. The dataset will be relevant not only for scientists but also to many stakeholders (water managers, policy-makers, NGOs). For all the above, this dataset could potentially become the new reference high quality dataset for precipitation for Spain. Therefore, I would recommend the publication of this paper after some minor revisions described below.

> Thank you for your kind comments. One of the aims of this dataset was to be clear and transparent. The inclusion of the uncertainty adds an extra value to the estimates of precipitation, showing their reliability and serving as a new element of climate analysis.

**SPECIFIC COMMENTS**
**I. The manuscript:**
The manuscript is generally well written, easy to follow and well structured. Below are a few specific comments to help improve some minor aspects of the paper.

1) In the introduction, page 1, line 27, the authors say "High-resolution spatiotemporal precipitation datasets are useful tools for land management", and that is the only example of potential application of their dataset they give. I can think of many more disciplines which would benefit from the existence of such a dataset, and I feel the introduction is not doing justice to the importance of this new dataset. The authors should emphasise much more on the value and impact this dataset will have in the scientific (and non-scientific) community for future studies in Spain. In the general comments above, I give a few examples of fields where such gridded meteorological datasets are in high demand. The authors could use that list as a starting point to expand their introduction. A reader should be more convinced on why this dataset is so important, the introduction should include some examples of applications with a few key publications illustrating that (for example from other countries where such dataset already exist). This dataset is really of high value, so this should be reflected in the introduction.

Thanks a lot for your suggestions that we have tried to follow to improve the introduction section.

**2) Page 2, line 14: Brunet et al., 2006. This paper was about gridded temperature dataset, so probably not too relevant here, unless you add the word "meteorological" before "datasets" in the sentence "a few daily datasets have been made for Spain or some of its regions".**

Right, that reference is about temperature but the paragraph was only about precipitation. The reference has been removed.

**3) Page 2, line 24-25: "The uncertainties of these estimations depend on the density of observations used to compute the model": you don't mention once in your paper the uncertainties coming from the uncertainties in the raw raingauge data. See comment below number 5.**

Thanks for the suggestion. We strongly agree with your comment about the uncertainties in rainfall measuring that we didn't consider. For this reason, and regarding your suggestions in comment #5, we try to highlight it now in the introduction section (page 3 lines 6-11).

*"However, unlike temperature, uncertainty in precipitation estimation is considerably higher. Firstly, raingauge measurements can be notoriously uncertain, especially in certain circumstances such as windy conditions (Rodda and Dixon, 2012) that can lead to significant undercatches in highly exposed areas (Rodda and Smith, 1986). Secondly, spatial variability of daily precipitation can be extremely high under certain atmospheric conditions such as convective processes that can occur at very local scale. Whereas the sum of these uncertainties is very difficult to estimate in practice, the uncertainties of these estimations are expected to decrease as the density of observations used to compute the models increase (Tveito et al., 2008; Hofstra et al., 2010). This issue has important implications in subsequent climatic analyses, and this is one consideration that needs to be taken into account (Beguería et al., 2015)."*

**4) Page 3, line 18-20: Please include the areas in km2 of Peninsular Spain, Balearic Islands and Canary Islands so that a non-Spanish reader gets a quick feel of the density of each network.**

The areas in km2 of the three geographical units have been included in section 2.

**5) Input data (page 3-4): Are the instruments uniform across the network? Which type of raingauge are they (automated or manual, or a mixture)? You do some quality control checks on your input data, which is very valuable. However, raingauge measurements are notoriously uncertain, especially in certain circumstances such as windy conditions (Rodda and Dixon, 2012), leading to significant undercatches, up to 16% in highly exposed areas (Rodda and Smith, 1986). Whereas these uncertainties are very difficult to estimate in practice, they should at least be acknowledged somewhere in your paper.**
**Rodda, J. C. and Dixon, H.: Rainfall measurement revisited, Weather, 67, 131-136, 2012.**
**Rodda, J. C. and Smith, S. W.: The significance of the systematic error in rainfall measurement for assessing wet deposition, Atmospheric Environment (1967), 20, 1059- 1064, 1986.**

Thank you for your interesting comments. The precipitation network comprises both automated and manual sensors. Although there can be differences between them as you indicate, the quality control criteria are applied to all types of data without distinction. Independently of the location or circumstances of each observation, all the data are treated equally. A drawback of this is that, effectively, we don't make differences between any type of data. However, the advantage is that, actually, this

doesn't matter because all the observations are compared with their surrounding stations and, when one is very different from the others (no matter why) it is removed. Anyway, we added an explanation about the origins of the data in section 2 to make it clear.

*"Most of the data were provided by the Spanish Meteorological Agency (AEMET), but we also used data from regional hydrological and meteorological services, and from the national agronomic network (Table 1). The greatest part of the information comes from manual stations, the automated ones entered service in mid 90s being in 2012 the 23% of the total in AEMET network."*

In addition, regarding the second part of your comment, please, see the response to the comment #3 where we address the uncertainty of rainfall measuring considering your references suggestion,

**6) Input data (page 3-4): When you measure precipitation, I assume this includes both rainfall and the water equivalent of new snow. It is probably worth stating that explicitly in the text.**

The variable extracted from the raw databases is "precipitation". Some offices as AEMET make a distinction between "precipitation" and "snow", but the rest only provide the information of precipitation. Some of these data can include the water equivalent of snow if the precipitation was solid in a specific circumstance. However, more than 90% of stations in Spain are under 1,500 m. a.s.l. which implies that the percent of snow days is a minimum fraction. Still, we added an explanation for this at the end of section 2:

*"Although the recovered information from raw databases of the meteorological offices was precipitation, in some cases this can include both rainfall and the water equivalent of snow if the source did not make the distinction."*

**7) Page 8 line 12: "greater radius": how much greater? If possible, it would be interesting to know the distance to the closest and the furthest 10 neighbours (maximum and mean) in Peninsular, Canary and Balearic Isles.**

The radius changes for each day and candidate location depending on the observations availability in its spatio-temporal environment. The function to estimate precipitation in the *reddPrec* R package includes a maximum threshold to find neighbouring stations that was set to 100 km. We added this in the text. Anyway, we appreciate your suggestion and we will consider the inclusion of an option to obtain this information in further versions of the R package. Moreover, we computed the average distances, their standard deviations and their minimum distances to the 10 nearest stations for IP, Balearic and Canary Islands and we added this to the text in page 4 lines 21-23.

*"The average distances to the 10 nearest stations were 24.67, 8.99 and 26.78 km for IP, Balearic and Canary Islands respectively with average standard deviations of 8.46, 3.69 and 17.32 km, and average minimum distances of 10.04, 3.01 and 5.51 respectively."*

**8) Page 9, line 10: "These months": which months are you referring to? November and December? Not clear from the text.**

Right, *"These months"* should be *"Summer months"*, it has been changed in the text. Thank you for the advice.

**9) Page 10: Analysis on NWD, CDDm and CWDm: very interesting. Although it would have been even more interesting if this was done per year or per decade which would have allowed to see the evolution over 60 years. It would be very nice if this could be included in the application example, but the authors might feel this goes beyond the scope of this paper.**

Thanks for the suggestion. We also think that a deeper study would be interesting in these extreme indices. Unfortunately, as you say this goes beyond the scope and the length of this paper. We intended to show the potential of the dataset to apply extreme precipitation indices. We are currently working in future papers about the recent changes in extreme precipitation.

**10) Page 12, line 16-17: see comments 3 and 5.**

As stated in previous responses, the uncertainty here is statistical. We used the error term of the precipitation modelling from the 10 nearest stations to assess the reliability of the predicted precipitation value (see page 7 line 20). We are aware about the uncertainties in precipitation measurement but, in this case, the QC process removes all the very-different data from the original dataset.

**11) Page 32, Table 5: How come MAE for the Peninsula has systematically higher values than in table 4?**

The calculations are different in both tables. Table 4 shows the errors of the averaged monthly sums of precipitation, which produces higher values than Table 5, which shows the mean daily errors. As the magnitudes of the observations are different (monthly and daily), the errors are also different.

**12) Page 33, Table 6: Please re-order the indices so that they are in the same order as in table 2. This would improve the readability greatly.**

We have changed the order to fit with Table 2. Thank you for the suggestion.

**II. The dataset:**
**The dataset was very easily downloadable, with adequate metadata. Ideally, it would have been good if the data complied with some existing international standards or conventions for NetCDF files, such as CF conventions. These conventions are developed to promote the interchange and sharing of netCDF files in a common format. Some tools or software require netcdf files to be CF or COARDS compliant. One recommendation of the CF conventions (which is compulsory for COARDS conventions) is that, for a multidimensional variable (such as the precipitation in this dataset), the order of the dimensions should be time, (height if present), lat (or y), lon (or x). This is not the case for this dataset, in which x comes before y, which means it can't be used with some tools. It is good practice to try and comply as much as possible to existing international standards so that datasets are as usable to as many users as possible. Having said that, this doesn't reduce the value of the dataset itself, which is still easily usable, and the order of the dimensions could be reasonably quickly fixed by the user if needed be.**

Thank you for your comments, you are right about the order of the dimensions in the NetCDF file, we will change it as soon as possible to make it compatible with all kind of software.

**Technical corrections:**
**Spelling: the use of English is inconsistent, the authors sometimes use British spelling (kilometre, neighbour, neighbouring, behaviour, favour, analyse) and sometimes American spelling (characterize, generalized, individualized, characterization, emphasizing). The authors should chose either British or American English and correct the spelling accordingly.**
**Page 3 line 11: remove the word "make" Page 15 line 6: replace "most" by "more"**

We aimed to complete the text in British English. We have improved the text and replaced the incorrect words. Thank you.